# Oxidative Stress: The Role of Antioxidant Phytochemicals in the Prevention and Treatment of Diseases

**DOI:** 10.3390/ijms25063264

**Published:** 2024-03-13

**Authors:** Adele Muscolo, Oliva Mariateresa, Torello Giulio, Russo Mariateresa

**Affiliations:** 1Department of AGRARIA, “Mediterranea” University, Feo di Vito, 89124 Reggio Calabria, Italy; mariateresa.oliva@unirc.it (O.M.); mariateresa.russo@unirc.it (R.M.); 2TL Pharma Via Lago di Martignano, 18, 65129 Pescara, Italy; giulio.torello@tlpharmaconsulting.it

**Keywords:** antioxidant phytochemicals, free radicals, chronic disease, health benefits, polyphenols

## Abstract

Oxidative stress, characterized by an imbalance favouring oxidants over antioxidants, is a key contributor to the development of various common diseases. Counteracting these oxidants is considered an effective strategy to mitigate the levels of oxidative stress in organisms. Numerous studies have indicated an inverse correlation between the consumption of vegetables and fruits and the risk of chronic diseases, attributing these health benefits to the presence of antioxidant phytochemicals in these foods. Phytochemicals, present in a wide range of foods and medicinal plants, play a pivotal role in preventing and treating chronic diseases induced by oxidative stress by working as antioxidants. These compounds exhibit potent antioxidant, anti-inflammatory, anti-aging, anticancer, and protective properties against cardiovascular diseases, diabetes mellitus, obesity, and neurodegenerative conditions. This comprehensive review delves into the significance of these compounds in averting and managing chronic diseases, elucidating the key sources of these invaluable elements. Additionally, it provides a summary of recent advancements in understanding the health benefits associated with antioxidant phytochemicals.

## 1. Introduction

External elements, coupled with internal contributors [1,2], collectively regulate the dynamics of reactive oxygen species (ROS). These molecular entities serve as primary second messengers, causing the activation of diverse signalling pathways that ultimately dictate cellular fate—whether it be mitogenesis or apoptosis [3]. ROS constitute a class of volatile molecules, such as hydrogen peroxide (H_2_O_2_), hydroxyl radical (OH^−^), singlet oxygen (1O^2^), and superoxide (O^2−^), produced by various cells [4]. The widespread distribution of ROS underscores their pivotal role in biosystems. While ROS are integral to pathogen resistance and cellular signalling, their status as potentially detrimental reactive entities is well established, inducing damage to intracellular proteins, lipids, and nucleic acids. This harm becomes particularly pronounced in disease mechanisms when ROS are not promptly scavenged [5].

Reactive oxygen species (ROS) generation occurs within environments characterized by elevated energy demands, necessitating a resilient metabolic framework. The dual nature of ROS, manifesting in both pathogenic and beneficial capacities associated with self-damage and the immune system, can be correlated with the energy demands inherent in these conditions.

While ROS are indispensable for essential biological functions, their constant generation can lead to a delicate equilibrium. An excess or imbalance between oxidants and antioxidants may give rise to a prevalent pathophysiological state known as oxidative stress [6]. Cumulative evidence underscores the intimate connection between heightened oxidative stress and a spectrum of chronic diseases, encompassing cardiovascular diseases, cancer, neurodegenerative diseases, diabetes, obesity, aging, and various chronic inflammatory conditions [7].

Excessive ROS production, prevalent in such conditions, results in oxidative damage, affecting both body and pathogens. ROS play a pervasive role in fundamental mechanisms and pathways, not only inflicting oxidative damage on cells and tissues but also actively participating in various homeostatic processes encompassing metabolism, immunity, growth, and differentiation [8]

The mitochondrial respiratory chain stands out as one of the primary contributors to cellular ROS, generating reactive oxygen species during ATP synthesis in normal oxygen metabolism. Consequently, ROS are commonly considered by-products resulting from the energy supply to cellular activities. An excessive production of ROS can result in oxidative harm to biomolecules, encompassing lipids, proteins, and DNA, contributing to the development of aging and various conditions such as cancer, respiratory issues, cardiovascular diseases, neurodegenerative disorders, and digestive ailments. The harmful effects of elevated ROS levels, known as oxidative stress (OS), have been linked to cell death [9]. To counteract the detrimental effects of OS, the body employs several mechanisms. Antioxidants, whether generated internally or supplied externally, play a crucial role in removing ROS, minimizing the oxidative stress [10]. Many phytochemicals act as antioxidants and are essential to neutralize OS. Phytochemicals are categorized into primary and secondary metabolites, depending on their roles in plant metabolism. Primary metabolites, essential for vegetal survival, encompass carbohydrates, amino acids, proteins, lipids, purines, and pyrimidines of nucleic acids. In contrast, secondary metabolites represent the rest of the chemical compounds generated from metabolic routes branching off the primary metabolic pathways [11]. Despite not having a direct impact on the growth, development, or reproduction of plants, they are able to enhance the ability of plants to thrive by facilitating the interaction and adaptation of plants to their environment.

While generally considered non-nutritive, the ability to mitigate chronic illnesses through the intake of fruits and vegetables is frequently ascribed to the presence of specific phytochemicals in these foods. These chemical constituents have been identified for their antiviral, antifungal, and antibiotic properties, acting as a defence mechanism against pathogens and safeguarding plants from severe leaf damage caused by UV light exposure [12]. Owing to their potent biological activities, for centuries, traditional medicine has harnessed the power of plant secondary metabolites, with the medicinal properties of plants frequently linked to these molecules [13]. Furthermore, different tissues and organs of medicinal plants may exhibit distinct medicinal properties at specific developmental phases. The accumulation of phytochemicals in medicinal plants is a response to the influences of different environmental factors such as the geographical locations of different altitudes, seasonal variation, and different types of soils [14]. Currently, these compounds are integral to various industries, including pharmaceuticals, cosmetics, and fine chemicals [15]. Phytochemicals with antioxidant properties are categorized into three primary sectors on the basis of their biosynthetic pathways (Figure 1): (a) nitrogen-containing compounds, including alkaloids, glucosinolates, and cyanogenic glycosides; (b) phenolic compounds, such as phenylpropanoids and flavonoids; and (c) terpenes [13,15]. Increasing evidence suggests that dietary phytochemicals go beyond simple antioxidant roles, influencing several cellular pathways linked to health and disease prevention [16]. These bioactive molecules, or their metabolites in the gut, interact with various biomolecules, particularly proteins, potentially affecting enzymes, cell receptors, or transcription factors. 

Their use has been linked to favourable impacts on physiologic processes by triggering transduction cascades linked to mitochondrial activity, inflammatory agents, epigenetic alterations, and the stimulation of endogenous antioxidant enzyme expression [17,18,19,20]. The notable variety in the structural composition of phytochemicals found in dietary sources renders them especially appealing for the drug discovery endeavour. The investigation into the preventive and therapeutic capabilities of phytochemicals has emerged as a pivotal focus of research. This review explores the significance of these compounds as important antioxidants in averting and managing chronic diseases while shedding light on key sources of these invaluable elements.

## 2. Origins of Phytochemicals with Antioxidant Properties: Exploring Their Source and Nature

A diverse array of phytochemicals is prevalent in fruits, vegetables, cereal grains, edible macrofungi, microalgae, and medicinal plants [21,22]. Traditional fruits like berries, grapes, Chinese dates, pomegranates, guavas, sweetsops, persimmons, Chinese wampees, and plums boast rich reservoirs of bioactive compounds [23,24]. Additionally, spontaneous fruits, such as those from *Eucalyptus robusta*, *Eurya nitida*, *Melastoma sanguineum*, *Melaleuca leucadendron*, *Lagerstroemia indica*, *Caryota mitis*, *Lagerstroemia speciosa*, and *Gordonia axillar*, have elevate antioxidant potential and a high amount of total phenolics [25]. Furthermore, discarded fruit parts (peels and seeds) are also rich in phytochemicals (Figure 2), including catechin, cyanidin 3-glucoside, epicatechin, gallic acid, kaempferol, and chlorogenic acid (Figure 3) [26,27]. Certain vegetables, such as cowpeas, allium cepa, sweet potato, green soybeans, pepper, ginseng, and broccoli, showcase elevated antioxidant power and amount of total phenolics [28,29]. In the realm of cereal grains, pigmented rice varieties like black rice, red rice, and purple rice stand out for their high levels of antioxidant phytochemicals, particularly flavones and tannins [30]. Additionally, various edible and wild flowers are recognized for their substantial content of antioxidant bio-compounds [31]. Polyphenols and carotenoids stand out as the primary categories of antioxidant biocompounds, significantly contributing to the antioxidant attributes inherent in various foods and plants. Notably, β-carotene, quercetin, myricetin, and kaempferol emerge as key antioxidant bioactive compounds identified in *Cape gooseberry* [32]. Meanwhile, strawberry boasts anthocyanins and ellagitannins as predominant phytochemicals within its profile [33]. Extracts from the pulp of *Euterpe oleracea* exhibit substantial antioxidant activity, attributed to the presence of flavonoids [34]. Within human diets, natural polyphenols rank as the most abundant antioxidants for their antioxidative potential, which is linked to the hydroxylation of aromatic rings in phenolics [35]. The concentration of polyphenols in food is subject to various influences, including cultivar, location, season, soil types, and conservation circumstances [36]. Dietary polyphenols encompass five distinct classes: flavonoids, phenolic acids, stilbenes, tannins, and coumarins. A further classification of flavonoids includes flavonols, flavones, flavanols, flavanones, anthocyanidins, and isoflavonoids [37]. The amount of polyphenols into the extracts of diverse fruits exhibits a positive linear association with total antioxidant activity. Fruits with higher total phenolic contents typically have a greater antioxidant activity [38]. For instance, the free radical neutralization of grape seed extract against ABTS radical is closely associated with the amount of phenols present [39]. Carotenoids, pigments contributing to warm hues in foods, comprise another group of essential phytochemicals. α-Carotene, β-carotene, lycopene, lutein, and cryptoxanthin represent the primary carotenoids present in both the diet and the human body (Figure 3). Fruits and vegetables serve as a primary source of carotenoids in the human diet. Notably, tomatoes are rich in lycopene, contributing to their characteristic red colour. Bergamot, in respect to other members of the citrus family, boasts a higher amount of polyphenols. 

These antioxidant compounds (Figure 4) may change numerous cellular processes, including mitochondrial function and SIRT pathways. An animal study by Ilari et al. indicates that bergamot polyphenols can restore mitochondrial functions and protect SIRT3 activity, potentially providing benefits in oxidative stress-triggered allodynia and hyperalgesia [40]. Raspberry ketones (RKs), phenolic compounds found in red raspberries, kiwifruit, peaches, and apples, have demonstrated hepatic, cardiovascular, and gastric protective properties in vitro and in vivo studies (Table 1). Mohamed et al.’s findings in this Special Issue suggest that RKs attenuate cyclophosphamide-induced pulmonary toxicity in mice by inhibiting oxidative stress and the nuclear factor kappa B (NF-κB) pathway [41].

Academic curiosity regarding dietary flavonoids has surged due to their postulated benefits. Chrysin, a flavonoid found in propolis, honey, passion fruit, and mushrooms, exhibited anti-hyperuricemic effects in a rat model nourished with a considerable amount of fructose corn syrup, as elucidated by Chang and colleagues. This effect was attributed to the antioxidant properties and deactivation of the inflammasome, leading to enhancements in conditions associated with metabolic diseases induced by hyperuricemia [58].

Flavonoids, recognized for their neuroprotective effects in diverse pathophysiological circumstances, could stimulate synaptogenesis and neurogenesis through the inhibition of oxidative stress and neuroinflammation. Cichon et al.’s thorough examination of the literature delves into the neurorestorative properties of flavonoids and their prospective role as catalysts for neuroplasticity in the management of central nervous system (CNS) diseases [59]. Additionally, a systematic review and meta-analysis by Ali et al. propose a noteworthy antidepressant impact of flavonoids in individuals exhibiting depressive symptoms [60]. Resveratrol, a polyphenol from the stilbene family that is not a flavonoid, has demonstrated potential advantages in the context of metabolic diseases. According to García-Martínez et al.’s systematic review and meta-analysis, resveratrol exhibits favourable effects on glucose concentration, insulin levels, and glycated hemoglobin (HbA1c) among individuals aged 45–59 years with type 2 diabetes mellitus (T2DM) [61].

However, its low bioavailability limits efficacy. Resveratrol butyrate ester (RBE), a novel derivative, exhibits increased biological activity, protecting against kidney damage and hypertension in a rat model of chronic kidney disease (CKD). Pterostilbene (PTS), a stilbenoid polyphenol with high bioavailability, shows promise in breast cancer cells by inducing epigenetic silencing of oncogenes. Harandi-Zadeh et al.’s experimental study provides new insights into the potential anticancer actions of PTS [62].

Carotenoids, fat-soluble plant pigments present in yellow-orange vegetables and fruits, have the potential to positively influence cognitive function for their antioxidant and anti-inflammatory properties. An analysis, drawing from data in nine intervention trials, proposes that supplementing with carotenoids may enhance cognitive performance among moderately healthy individuals aged 45–78 years [20]. *Seseli L*. species (*S. gummiferum* and *S. transcaucasicum*), renowned for their traditional utilization in herbal remedies, contain high amount of polyphenolic compounds, displaying antioxidant effects, bactericidal activity, and inhibitory effects on enzymes associated with several diseases [63]. Spirulina platensis, a photosynthetic alga, possesses elevated concentrations of antioxidants, notably phycocyanin (PC). The research conducted by Omar et al. on broiler chickens suggests that PC derived from S. platensis has the potential to function as a natural growth promoter, antioxidant, and anti-inflammatory additive in feed [64]. Phytoestrogens, classified into five main classes, have shown high potential in preventing chronic diseases, modulating epigenetic processes [65]. *Withania somnifera* (Indian ginseng) contains over 35 bioactive phytochemicals, with withaferin A (WA) recognized as a powerful anti-cancer and anti-inflammatory agent, with poly-pharmacological mechanisms of action that substantiate its potential in addressing diverse chronic inflammatory diseases [66]. Due to its abundant array of phytochemicals known for their health-promoting benefits [67], pomegranate fruit has earned the title of the foremost member in the *Super Fruits* category. The pharmaceutical industry also incorporates extracts from pomegranate for the creation of capsule supplements [68]. Globally, pomegranate cultivation encompasses around 300,000 hectares, yielding an impressive three million tons of production [69]. Surprisingly, Marra et al. [27] evidenced that the waste products of pomegranates, as well as other fruits classified among those containing a high quantity of bioactive compounds, are the parts that contain the most biocompounds. This is because these biocompounds, including antibacterial, antibiotic, and antioxidant properties, have the ability to protect the fruit from external parasite attacks. In their works, Marra et al. [27] highlighted that pomegranate peel is a byproduct rich in bioactive compounds. The quantity and variety of these bioactive compounds are contingent upon the cultivar as well as the geographical region in which the plant is cultivated. Cultivars of the same type, when grown under different conditions, exhibit varying phenolic content and diverse antioxidant activities. In pomegranate peels of Wonderful cultivars grown in Calabria, total phenolic substances, total flavonoids, vitamin C, vitamin E, and antioxidant activities, as well as single phenolic acids and flavonoids were much higher than other cultivars and also Wonderful plants cultivated in other locations worldwide.

## 3. Bioavailability of Phytochemicals

The notion of bioavailability emerged as a means to measure the absorption, distribution, metabolism, and eventual excretion of micronutrients and phytochemicals. It aims to quantify the amount of these compounds that are effectively utilized within the body. Defined as “the rate and extent at which the therapeutic entity is absorbed and made available at the site of drug action” [70], bioavailability delineates the efficiency of absorption and utilization. The journey of orally ingested phytochemicals within the biological system encompasses a series of sequential processes, encompassing digestion, liberation, solubilization, absorption, distribution, metabolism, and eventual excretion. Dynamic interactions of a physical, chemical, and biological nature transpire within the gastrointestinal tract (GIT), facilitating the liberation of phytochemicals from the consumed food matrices. Mechanical forces contribute to a reduction in particle size while the acidic gastric fluids facilitate the breakdown of distinct food structures. Digestive enzymes play a pivotal role in the hydrolysis of fats, proteins, and carbohydrates, while bile salts aid in solubilizing and transporting lipids. Upon release from the food matrix, phytochemicals may undergo absorption by the cell layer lining the GIT via passive or active transport mechanisms. Subsequently, these phytochemicals may embark on a journey to the bloodstream through either the portal vein (for hydrophilic compounds) or the lymphatic system (for hydrophobic compounds), contingent upon their polarity.

Once in the bloodstream, these phytochemicals traverse the systemic circulation system, reaching various tissues and organs throughout the body. Notably, any unutilized phytochemicals and their metabolites may undergo excretion, exiting the body through either urine or faeces. This intricate series of processes underscores the intricate fate of orally ingested phytochemicals within the biological system. In a study involving fifteen healthy volunteers, the consumption of 144 g of raisins led to the identification and quantification of seventeen phytochemicals, including sixteen phenolics and oleanolic acid, in the volunteers’ plasma, underscoring the bioavailability of phytochemicals in raisins [71]. Additionally, an eight-week intake of soy germ-fortified tomato juice, providing 66 mg isoflavones and 22 mg lycopene, resulted in a notable change in plasma lycopene concentration from 0.60 ± 0.22 to 1.24 ± 0.30 μmol/L in eighteen healthy men and women. This intake saw a substantial increase in several health markers, including resistance to oxidation of LDL + VLDL-C, HDL-C levels, and the total-C/HDL-C ratio [72]. Similar positive outcomes were also observed in older adults [73].

However, certain phytochemicals encounter limitations in solubility, stability, bioavailability, and target specificity within the body, making them less able to attain effective levels in target tissues. This is notably the case with compounds like EGCG, resveratrol, curcumin, and quercetin [74]. For instance, EGCG showed low bioavailability, with oral administration resulting in an area under the concentration–time curve (AUC) of 39.6 ± 14.2 μg·h/L compared to 2772 ± 480 μg·h/L via intravenous administration (10 mg/kg). This disparity points to the rapid degradation of EGCG in body fluids [75]. Similarly, quercetin’s low aqueous solubility and rapid metabolism in the body reduce its effectiveness in disease prevention and treatment [76]. Studies indicate that quercetin’s Cmax was 0.16 μM after ingesting grape juice containing 10 mg quercetin aglycone, representing a mere 1.4% of the ingested dose [77]. 

Resveratrol also demonstrated limited oral bioavailability, with maximal plasma concentrations around 10 ng/mL following an oral dose of 25 mg. However, its concentration and that of total metabolites were significantly higher, at around 400–500 ng/mL, evidencing a scarce bioavailability of free resveratrol [77]. Similarly, curcumin exhibited exceedingly low serum levels in different doses, with 4 mg, 6 mg, and 8 mg/day dosages yielding average peak serum concentrations of 0.51 ± 0.11 μM, 0.63 ± 0.06 μM, and 1.77 ± 1.87 μM, respectively. Strategies involving co-administration with piperine, phytosome technology, and nanoparticles have been explored to improve curcumin bioavailability, resulting in substantial enhancement in both human and rat studies [78]. For instance, the co-administration of 2 g of curcumin with 20 mg of piperine enhanced curcumin bioavailability by 20 times in humans and 1.56 times in rats [79]. Nanoparticles and innovative delivery systems like PHYTOSOME^®^ have also been explored to enhance the assimilation and bioavailability of polyphenolics [80]. 

The bioavailability and bioactivity of phytochemicals are intricately linked to the chemical composition of the food matrix and the processing operations employed [81,82,83]. The ingestion of hydrophobic phytochemicals enhances their bioaccessibility [84]; conversely, the occurrence of dietary fibres impedes the release of phytochemicals from the food matrix, reducing their bioaccessibility, though exceptions exist such as certain dietary fibres, like pectin, that can enhance bioaccessibility [85,86]. Additionally, processing operations exert complex effects on phytochemical bioaccessibility. Thermal processing can either decrease or increase bioavailability by promoting chemical degradation or enhancing liberation from the food substrate [87,88]. The integration of heat treatments with different methods as microwave, ultrasound, or vacuum techniques, showed promise in mitigating negative impacts on phytochemicals induced by heating [89,90]. Non-thermal technologies like freeze-drying, high-pressure processing, pulsed electric field, and bioprocessing have also demonstrated efficacy in reducing degradation and enhancing bioaccessibility [91,92]. The combined impact of these factors can lead to significant inter-individual and intra-individual changes in bioavailability, occasionally varying from 0% to 100% of the consumed quantity.

## 4. Bioaccessibility of Phytochemicals

Recently, researchers have explored chemical and physical modifications to improve the bioaccessibility of phytochemicals. Chemical modifications enhance stability, solubility, and bioaccessibility, though safety concerns are a consideration [93]. Physical approaches, often preferred for safety reasons, include encapsulation into particle-based carriers assembled from edible biopolymers or lipids [94]. Encapsulation has demonstrated improvements in dispersibility, stability, bioaccessibility, bioavailability, and regulated discharge patterns for phytochemicals [95].

Diverse models, including in vitro digestion simulation, cellular uptake, and in vivo studies, have been employed to assess the biological destiny of phytochemicals taken orally. In vitro digestion models, being cost-effective and reproducible, are frequently used for assessing bioaccessibility [96,97]. Zhang et al. [98] offered insight into overall bioavailability, evidencing that cellular uptake mimics absorption from the gastrointestinal tract. In vivo animal studies provided more accurate predictions but were limited by high costs, extended timelines, and ethical concerns [99,100]. Recent human studies have contributed valuable insights [101,102], though challenges associated with in vivo studies underscore the need for more thorough and dependable in vitro techniques (Table 2). Due to the diverse chemical structures found in nutraceuticals, their bioaccessibility and bioavailability were subject to fluctuations based on specific chemical and physio-chemical parameters. In the evaluation of bioaccessibility, one can potentially adjust various parameters to gauge oral efficiency, such as pH and temperature ranges, along with enzyme activity. Nevertheless, the kinetics of the absorption process may also be influenced not solely by the mentioned factors but also by the presence of other foods that can exert an additive or antagonistic or synergistic effect. 

Highlighting the significance of bioaccessibility in gauging the bioavailability of nutraceuticals and functional foods underscores the impact of external factors, that can be manipulated to improve the nutritional benefit and quality of these products. Noteworthy influential external parameters encompass (a) the chemical and physical properties of the nutraceutical product, (b) the integration of innovative delivery systems, and (c) the processing and storage conditions of the product [114]. Specifically, for nutraceuticals, the composition, dose, and structure can wield substantial influence over the ultimate bioavailability of phytonutrients derived from the matrix. In a typical diet, humans ingest several grams per day of phytochemicals. However, due to the diverse array of structurally distinct compounds and the modest uptake and bioavailability of certain phytochemicals, their presence as either parent compounds or metabolites at the systemic and tissue levels remains relatively scarce, typically in the micromolar range. The diminished bioavailability of specific phytochemicals, particularly when compared to macronutrients, stems from their recognition and processing by the body as xenobiotics. Consequently, the estimated absorption of polyphenols, based on urinary data from healthy volunteers and studies involving individuals with ileostomies, ranged between 1% and 60% [115].

Nicoleșcu et al. [116] emphasized the crucial role of the bioaccessibility of carotenoids, vitamins A, D, E, and K, and longer chain fatty acids. Bioaccessibility became the rate-limiting step due to the necessity of forming mixed micelles for solubilization.

The significance of bioaccessibility in assessing the bioavailability of nutraceuticals and functional foods is emphasized by external factors that impact their nutritional value and quality. Prominent external parameters encompass the chemical and physical attributes of the nutraceutical product, the implementation of novel delivery systems, and the processing and storage conditions [114]. In the context of nutraceuticals, the dosage form plays a crucial role in influencing the ultimate bioavailability of phytonutrients from the matrix. Specific excipients, such as propylene glycol solutions, phospholipid complexes, nanoparticles, and diverse colloidal systems, significantly modify the bioavailability in this regard, and can enhance accessibility to intestinal absorption [117].

Predicting the behaviour of nutraceuticals with a diverse range of phytochemicals poses challenges, considering the need to account for each individual physical and chemical property. Hydrophobicity correlates with lower solubility in gastrointestinal fluids, leading to decreased bioaccessibility, while hydrophilic behaviour is associated with elevated solubility but scarce permeability through the epithelial wall [114]. For compounds like polyphenol esters and glycosides, characterized by low solubility and absorption, enzymatic hydrolysis is required for absorption as aglycones, mainly facilitated by bacterial enzymes [118].

Thakur et al. [115] recently identified processing techniques’ influence on phytonutrient bioaccessibility in various functional foods. Cooking processes, particularly those inducing cell wall rupture, enhance the release of polyphenolic compounds and carotenoids. Dehydration, thermal processing, drying, frying, and the addition of oils and fats also contribute to increased matrix release compared to raw products. Non-thermal processing technologies like ultrasound, pulsed electric field, and high pressure can enhance bioaccessibility by promoting cell membrane permeability, but they may lead to higher viscosity due to fibre and pectin release, potentially impacting bioaccessibility [119]. 

The impact of thermal and non-thermal processing technologies on bioaccessibility varies in respect to phytochemical type, plant, or food substrate. These technologies may either increase or decrease bioaccessibility, with levels influenced not only by the selected method but also by pre-treatment steps and the nature of the compounds under investigation [120]. The absorption can be impacted by antinutrients, such as phytates, polyphenols, and dietary fibres, that can impede the absorption process, diminishing bioaccessibility and negatively affecting the bioavailability of minerals and micronutrients [121]. A significant challenge associated with the bioaccessibility of phytochemicals is intricately linked to the microbiota and consequently intestinal pH. Recent research indicates that the gut microbiota enhances the bioactivity of phytochemicals and serves as a symbiotic partner [122]. The colonic microbiota plays a pivotal role in metabolic homeostasis, contributing to the gut–brain axis by producing neurotransmitters and metabolites like serotonin and γ-aminobutyric acid (GABA). These compounds modulate emotions, behaviour, neuronal signalling, digestive function, and the immune system of the host [123]. Dysbiosis in pregnant mice has been shown to seriously impair foetal neurodevelopment. The colonic microbiota actively participates in the absorption of phytochemicals, breaking down undigested polyphenols through the production of specific enzymes responsible for processes such as polyphenol deglycosylation, demethylation, dehydroxylation, ester cleavage, isomerization, ring fission, and decarboxylation [124].

Various sets of colonic microbiota cooperate to metabolize different phytochemicals. For instance, ellagic acids are metabolized to urolithins by bacteria like *Clostridium* spp., *Ruminococcaceae*, *Eubacterium* spp., *Gordonibacter* spp., and *Ellagibacter isourolithinifaciens* [125]. Daidzeins, isoflavones from soybeans, are instead converted into equols by another community of bacteria such as *Streptococcus intermedius*, *Bacteroides ovatus*, and *Ruminococcus productus* [126]. Lignans undergo modifications by diverse bacteria, including *Clostridium scindens*, *Eggerthella lenta*, *Clostridiales*, and *Lactonifactor longoviformis* [126]. Consequently, phytochemical metabolites can vary among individuals consuming the same food due to the individual differences in colonic microbiota composition. Many diseases heavily impacted by dietary factors entail oxidative damage as an initial occurrence or an early stage in the progression of the disease. Consequently, a significant emphasis in dietary disease prevention has been placed on antioxidant intervention. Over the past decade, a wealth of research, including numerous human intervention studies, has consistently highlighted the pivotal role of antioxidants, particularly phytochemicals, in mitigating the risk of chronic diseases [127].

Traditionally, the beneficial role of antioxidants has been associated with curtailing the undesirable and uncontrolled production of reactive oxygen species, leading to a state known as oxidative stress. However, contemporary scientific understanding increasingly recognizes that the mechanism of action of antioxidants in vivo may be far more intricate than previously thought.

Beyond their role in mitigating oxidative stress, antioxidants and phytochemicals demonstrate multifaceted mechanisms that contribute to disease prevention. These mechanisms include the modulation of inflammatory pathways, the enhancement of cellular repair and regeneration, and interaction with signalling cascades involved in cell growth and apoptosis. Moreover, antioxidants exhibit the potential to influence epigenetic processes, altering gene expression patterns associated with disease susceptibility. Recent studies have illuminated the intricate interplay between antioxidants and the gut microbiota, revealing a symbiotic relationship. Antioxidants, particularly those derived from plant-based sources, can impact the amount and diversity of the gut microbiota, which, in turn, contribute to overall health and disease prevention. Additionally, antioxidants demonstrate neuroprotective effects and may play a crucial role in preserving cognitive function and preventing neurodegenerative diseases.

As research progresses, the understanding of the comprehensive impact of antioxidants and phytochemicals on human health continues to evolve. Insights into their nuanced mechanisms of action open avenues for targeted interventions, personalized nutrition strategies, and the development of novel therapeutic approaches for a wide array of diseases influenced by dietary factors.

## 5. Phytochemicals in the Prevention and Treatment of Human Disease

The safeguarding effect that phytochemicals such as tannins, flavones, triterpenoids, steroids, saponins, and alkaloids exert against the onset of chronic diseases may be linked to their antioxidant activity, countering the harmful effects of overproduced oxidants such as reactive oxygen species and reactive nitrogen species. Historically, pharmaceutical companies have traditionally explored natural plant products as a primary reservoir for the discovery of new drugs. In recent years, numerous epidemiologic and case–control studies have delved into mechanistic interactions, shedding light on the multifaceted benefits of polyphenols. Both animal and human studies have consistently demonstrated that the anti-inflammatory and antioxidant properties of polyphenols lay a solid foundation for their potential role in preventing and treating various non-communicable diseases (NCDs). These diseases encompass a broad spectrum, including but not limited to carcinogens, cardiovascular diseases (CVDs), diabetes, pancreatitis, osteoporosis, gastrointestinal dysfunctions, lung damage, and neurodegenerative diseases.

Current research indicates that the sustained consumption of polyphenols over time can offer protection against the onset and progression of such health conditions [128]. Notably, polyphenols have emerged as promising agents in the mitigation of oxidative stress, particularly evident in in vitro studies showcasing their efficacy in countering oxidative stress in HepG2 cells.

The growing body of evidence supporting the diverse therapeutic effects of polyphenols underscores their potential significance in holistic health management. A further exploration of their mechanisms and clinical applications holds promise for enhancing our understanding of their preventive and therapeutic roles in combating a range of prevalent and debilitating diseases. In detail, the protective potential of polyphenols against oxidative stress arises from their capacity to generate hydrogen peroxide, a molecule known for its regulatory influence on immune responses and cellular growth [129]. However, it is noteworthy that certain negative effects have been reported in individuals with degenerative diseases such as high blood pressure, thyroid disorders, epilepsy, or heart diseases, attributed to pre-absorptive interactions during digestion [130]. 

Numerous studies have elucidated that dietary polyphenols play a crucial role in inhibiting proinflammatory transcription factors by interacting with proteins responsible for gene expression or cell signalling. This interaction serves as a preventive measure against several chronic diseases mediated by inflammation [131]. Specifically, the therapeutic potential of anthocyanins in addressing type 2 diabetes in both animals and humans is grounded in the polyphenolic compounds’ ability to safeguard beta cells from oxidation. Furthermore, these compounds exhibit noteworthy anti-inflammatory and antioxidant actions, resulting in a reduction in starch digestion. The cumulative effect underscores the promise of polyphenols, particularly anthocyanins, in preventing and treating various health conditions associated with inflammation and oxidative stress. Polyphenols stand out as the predominant antioxidants in the human diet, surpassing the intake of vitamin C by a factor of 10 and vitamin E by 20 times [132]. Extensively investigated by numerous researchers, polyphenols have garnered attention for their potential impact on cardiovascular diseases (CVD), showcasing positive outcomes. A consensus emerges from various studies, indicating that the flavonoid class of dietary polyphenols plays a pivotal role in reducing the incidence of CVDs [133]. Notably, specific subclasses of polyphenols, including anthocyanidins, flavan-3-ols, flavones, flavonols, and proanthocyanidins, have been associated with a decrease in cardiovascular risk [107,133]. Among flavonoids, apigenin and luteolin, both flavones, were associated with antioxidant activity. Luteolin, in fact, exhibited beneficial characteristics, including anti-inflammatory and antibacterial activities [134]. Apigenin, on the other hand, inhibited the tumour-promoting effects of 12-O-tetradecanoylphorbol-1,3-acetate (TPA) on mouse skin, similar to curcumin, a dietary pigmented polyphenol. This effect is potentially achieved through the suppression of protein kinase C activity and nuclear oncogene expression [133]. Apigenin also boasts antibacterial, anti-inflammatory, diuretic, and hypotensive properties, and promotes smooth muscle relaxation [134]. Another flavonoid, myricetin, a hexahydroxyflavone, exhibits antibacterial activity and anti-gonadotropic effects [134]. Flavonoids exhibit potential brain protection through various mechanisms, including shielding vulnerable neurons, increasing the neuronal function, and stimulating neuronal regeneration [135]. In the realm of Parkinson’s disease, the citrus flavanone tangeretin showed promise in maintaining nigrostriatal integrity and preserving performance characteristics following 6-hydroxydopamine lesioning. This suggests its potential as a therapeutic agent against the underlying pathology associated with Parkinson’s disease. Beyond flavonoids, phenolic substances such as cinnamic acid and tyrosol have demonstrated neuroprotective effects by safeguarding against 5-S-cysteinyl-dopamine and peroxynitrite neurotoxicity in vitro. Resveratrol, a natural polyphenol belonging to the stilbene family and derived from wine, boasts robust antioxidant, anti-inflammatory, and anti-aging properties. Research underscores its protective effects against several cardiovascular diseases [136].

The substantial prevalence of polyphenols in the human diet, coupled with their demonstrated positive impact on cardiovascular health, accentuates their potential as valuable dietary components. The nuanced understanding of specific subclasses, such as flavonoids and resveratrol, contributes to a more targeted approach in harnessing the cardiovascular benefits offered by these natural compounds. Over the past decade, a wealth of preclinical, clinical, and epidemiological investigations has underscored the potential of natural compounds, particularly polyphenols, in both the treatment and prevention of cancer. The dynamic interplay between polyphenol consumption and cancer incidence has been the subject of consistent scrutiny, with frequent meta-analyses offering valuable insights. In these meta-analyses, resveratrol has been the subject of extensive study for its anticancer properties across various malignancies, including skin, breast, prostate, gastrointestinal, and lung cancers. Notably, its primary anticancer mechanisms are believed to involve the modification of genetic and epigenetic variables [137]. Resveratrol intake becomes paramount for maintaining a careful balance in regulating oestrogen levels for the prevention of breast cancer. Oestrogen hormone therapy administered to menopausal women significantly increases the risk of breast cancer. Resveratrol, through the activation of the Nrf2 gene expression, demonstrates the ability to enhance the expression of UGT1A8, leading to the degradation of catechol oestrogen. Notably, Nrf2 influences the promoter of the UGT1A8 gene, ultimately inducing its activation [138]. This intricate interplay emphasizes the potential of resveratrol in modulating oestrogen metabolism and offers a nuanced approach to mitigating the carcinogenic effects associated with oestrogen, particularly in the context of breast cancer prevention. It is also worth mentioning that resveratrol (RSV) acts as a suppressor of the mitogen-activated protein kinase (MAPK/NF-kB) pathway that is involved in tumour migration and invasion [139]. Hence, this phytochemical exhibits potential protective qualities against coronary heart disease and demonstrates therapeutic activity against cancer. Resveratrol playing a crucial role as an antioxidant is considered a promising therapeutic agent in the management of cancer. To harness its protective effects, incorporating foods rich in resveratrol (RSV) into the diet, such as grapes, strawberries, blueberries, peanuts, and cocoa, is highly recommended. Enzymes located in the gut extensively metabolize RSV, leading to low oral bioavailability due to pre-systemic elimination. Over the past decade, diverse methodological approaches, including encapsulation, lipid nanocarriers, and emulsions, have been devised to enhance the limited bioavailability of RSV [140]. Ovarian cancer, often detected at advanced stages, has seen potential mitigation through the consumption of isoflavone that emerged as a standout contributor, associated with a significant reduction (19%) in the risk of gastric cancer [141]. Meanwhile, Moussavi et al. [142] delved into the impact of apigenin, a dietary component, on colon cancer, exploring its association with cell death mediated by ROS. Epidemiological studies have also shed light on the role of soy product consumption, proving its capacity to lower the incidence rate of breast cancer [143]. Additionally, pure soy isoflavones, including genistein, genistin, daidzein, and biochanin A, along with a concentrated soy phytochemical, demonstrated growth inhibition that was dependent on the dosage in human (HT1376, UMUC3, RT4, J82, and TCCSUP) bladder cancer cell lines. The extent of inhibition varied based on the specific cell line under consideration [144]. Furthermore, soy isoflavones induced a G1M cell cycle arrest across all evaluated human cell lines, as determined using flow cytometry. In some instances, certain cancer cell lines exhibited DNA fragmentation consistent with apoptosis.

What it is important to know is that the gut microbiota play crucial roles in the transformation of polyphenolic compounds by influencing normal biochemical features and contributing to variations in the response to polyphenolic treatments among individuals. Polyphenolic compounds found in nature are complex molecules with scarce solubility and limited bioavailability [145]. When introduced into the human gastrointestinal tract (GIT) through dietary intake or supplements, these compounds undergo transformations facilitated by enzymes and microbiota, resulting in enhanced bioavailability and increased pharmacologic properties [146]. A recent study demonstrated that gut microbiota can convert ellagic acid, a phenolic acid from berries, into urolithin. This transformed urolithin exhibits a potent inhibition of heme peroxidases, including myeloperoxidase and lactoperoxidase, leading to a significant reduction in inflammation-mediated cellular destruction [147]. Additionally, other studies revealed the production of soluble polyphenols such as ferulic acid (FA) and coumaric acid through the hydrolytic activity of gut microbiota on ester-linked arabinoxylans [148]. The gut microbiota have been observed to degrade ferulic acid dimers into vanillin and 3-(4-hydroxyphenyl)-propionic acid, showcasing enhanced anticancer activity. The regular consumption of dietary polyphenols is associated with various health-promoting effects. The reported structural composition and interactions with other food compounds play a pivotal role in influencing the activities and bioavailability of polyphenols, thereby causing diverse biological effects. The distinct structures of polyphenols contribute to their unique antioxidant features. Additionally, research suggests that when different polyphenols are combined with various food compounds, their interaction may exhibit an additive, synergistic, or antagonistic effect on their diverse bioactive compounds [149]. The interplay between polyphenols and proteins can lead to the formation of soluble or insoluble complexes, thereby influencing the bioavailability of these bioactive compounds. The complex formations between polyphenols and proteins have the potential to either reduce or enhance the antioxidant capabilities of dietary polyphenols. Polyphenols interact with various carbohydrates originating from the cell wall, like pectin, cellulose, or dietary fibres. Several consequences may arise from these interactions. Firstly, there is an impact on the bioavailability of phenolic compounds. Studies have indicated an influence on the processing properties of carbohydrates. There is a reported decrease in bioavailability due to the formation of associations between carbohydrates and polyphenols, wherein polyphenols become captured within the carbohydrate structure. The Bioavailability Index, calculated at the intestinal level [150,151], of the most representative phytochemicals with antioxidant activity is reported in Table 3.

## 6. Discussion

There is robust evidence supporting the positive impact of specific polyphenols in preventing and also treating cardiovascular diseases, neurodegenerative disorders, cancers, and complications related to obesity. However, our ability to fully harness these benefits is constrained by our current understanding of interactive mechanisms, optimal dosage requirements, and potential side effects.

Polyphenols coexist within complex food environments, often interacting with surrounding food compounds. These interactions extend to lipids, carbohydrates, proteins, and other micro food components in human diets, playing roles that could be pivotal. Recent studies highlight the significance of a balanced incorporation of polyphenols into diets, emphasizing their high antioxidant and bioactive properties essential for maintaining good health and preventing prevalent human NCDs.

Several studies have elucidated the antioxidant and anti-inflammatory effects of polyphenols derived from various sources in diverse in vitro and in vivo experimental settings. Maximizing the potential benefits of dietary polyphenols may hinge on a comprehensive understanding of how they interact with each other, pointing toward the need for further exploration and refinement in this field. Thus, emerging plant-based nutraceuticals may become recognized as an essential element of disease-preventive dietary components. Exciting opportunities await the food industry in crafting innovative food products through the future development of nutraceuticals from both plant and animal sources.

## 7. Methods

In meticulously crafting this article’s literature survey, we extensively relied on esteemed scientific resources such as PubMed, ScienceDirect, Scopus, ResearchGate, and Google Scholar, with a deliberate emphasis on the comprehensive databases of ScienceDirect and Scopus. This review encompasses recent articles that delve into various pharma-nutritional aspects of antioxidant phytochemicals, with a particular focus on the fundamental characteristics of ROS, sources of antioxidant phytochemicals, and their involvement in the prevention and therapy of several long-lasting metabolic disorders. Understanding the intricate interplay of ROS in cellular processes may uncover novel insights into their nuanced functions and potential therapeutic interventions. The keywords employed for study collection were meticulously chosen and included “biotic stress”, “plant secondary metabolites”, “defense mechanism, antioxidants, phytochemicals, reactive oxygen species, antioxidants in prevention and treatments of human disease”. The paramount focus was centred on discerningly selecting seminal works that uniquely contributed to the in-depth exploration of the topics covered, ensuring a rigorous and thorough examination of all pertinent information embedded within the chosen articles.

## Figures and Tables

**Figure 1 ijms-25-03264-f001:**
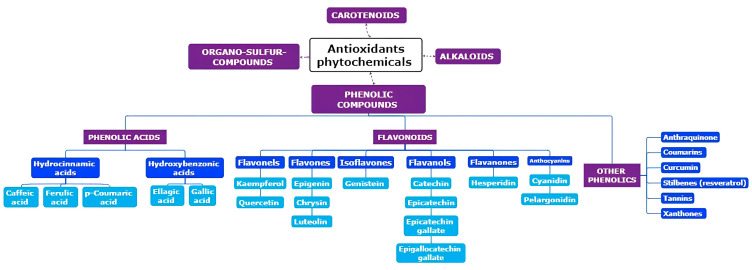
Classification of dietary antioxidant phytochemicals.

**Figure 2 ijms-25-03264-f002:**
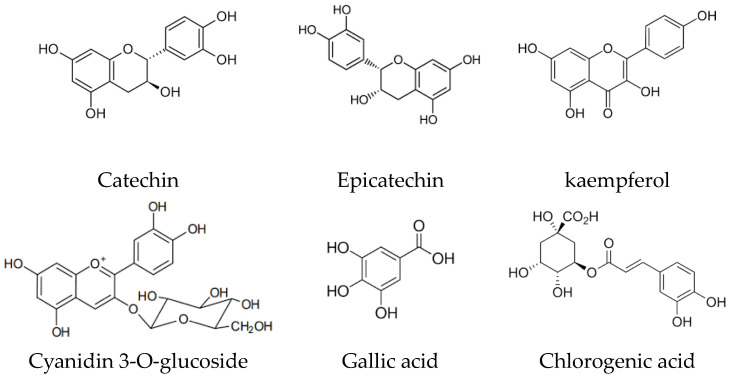
Chemical structures of some phytochemical polyphenol antioxidants featuring a basic phenolic (benzene) ring with two or more hydroxyl (OH) groups. These bioactive compounds act as antioxidants and are recognized as modulators of epigenetic gene expression regulation.

**Figure 3 ijms-25-03264-f003:**
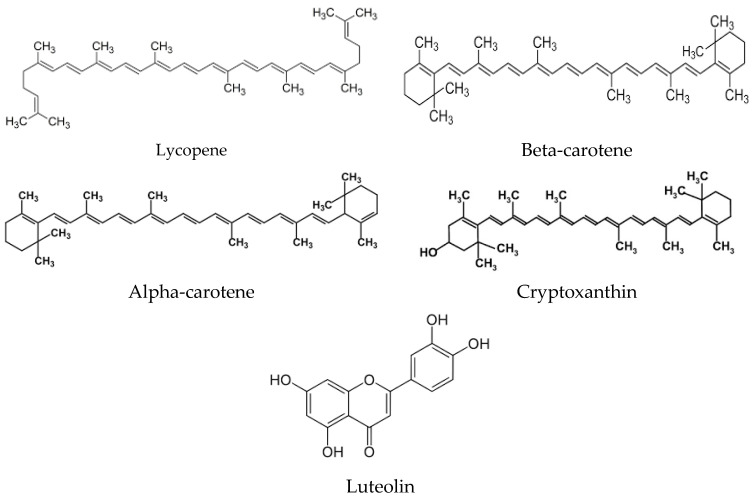
Chemical structures representative of the most active carotenoids present in fruits and vegetables, and in the human body.

**Figure 4 ijms-25-03264-f004:**
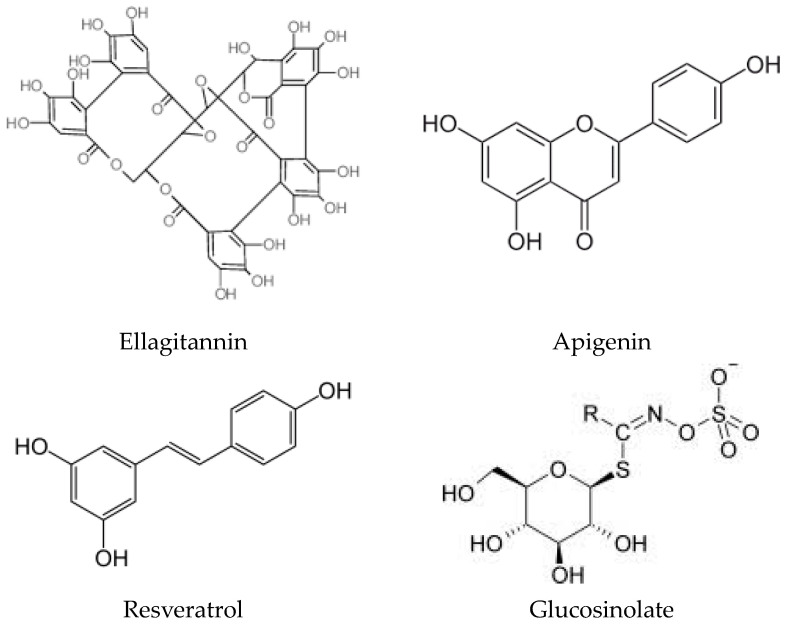
Chemical structure of polyphenols with protective effects and health benefits.

**Table 1 ijms-25-03264-t001:** Protective effects of polyphenols extracted from different food matrices: in vitro study.

Type of Polyphenol	Target	Mechanism of Action	Modification Types	Reference
Epigallocatechin gallate	Lung Cancer	Causes a decrease of Hepatoma-derived growth factor (HDGF)	Decreased tumour multiplicity in mice	[42]
Epigallocatechin gallate	Colorectal Cancer	Causes inhibition of cell proliferation induction of Nrf2 nuclear translocation and autophagy, expression of LC3 and caspase-9 mRNA	Reduced tumour multiplicity, tumour size	[43]
Epigallocatechin gallate	Skin Cancer	Causes inhibition of the proliferation, inhibition of NF-κB activity, IL-1β secretion related with downregulation of NLRP1	Inhibited melanoma tumour growth	[44]
Epigallocatechin gallate	Prostate Cancer	Causes inhibition of class I HDACs (HDAC1, 2, 3, and 8), induction of tumour cell apoptosis	Inhibited tumour growth	[45]
Epigallocatechin gallate	Breast Cancer	Causes inhibition of cell growth	Inhibited tumour growth	[46]
Epigallocatechin gallate	Breast Cancer	Modifies activation of caspases-3, -8, and -9	Inhibited tumour growth	[47]
Epigallocatechin gallate	Breast Cancer	Causes decrease of AKT and increase Bax/Bcl-2 ratio, comparable to tamoxifen	Inhibited tumour growth	[48]
Ellagitannin	Leukaemia	Causes apoptosis	Inhibited proliferation of leukemic cells	[49]
Glucosinolate	Liver Cancer	Causes apoptosis	Reduced tumour growth	[50]
Resveratrol	αVβ3 integrin receptor in MCF-7 cells	Causes apoptosis of breast cancer cells	Reduced tumour growth	[51]
Resveratrol	Liver Cancer	Enhances autophagic flux and apoptosis simultaneously in a dose- and time-dependent manner in HL-60 cells and Hepa 1c1c7 cells	Cancer chemo-preventive agent	[52]
Resveratrol	Human Prostate Cancer	Causes a decrease in DU-145, PC-3, and JCA- levels	Decreased prostate cancer cell growth	[53]
Apigenin	Glycaemia	Causes an increase in the activity of cellular antioxidants, catalase, superoxide dismutase and glutathione.	Decreased and prevented hyperglycaemia	[54]
(–)-Epicatechin	Breast Cancer	Causes inhibition of the MCF-7 cell viability	Decreased tumour cell growth	[55]
Luteolin	Breast Cancer	Causes decrease in the viability of MCF-7 breast cancer cells	Decreased breast cancer cell growth	[56]
Gallic acid	Colon Cancer (rats)	Causes an increase in superoxide dismutase, catalase, glutathione reductase, and glutathione peroxidase activities	Decreased widespread cancer	[57]

Abbreviations: Nrf2 (nuclear factor erythroid 2–related factor 2); LC3 (Light Chain 3); NLPR1 (NLR Family Pyrin Domain Containing 1); HDACs (Histone deacetylases); AKT (protein kinase B); Bcl-2 (B-cell lymphoma protein 2; Bax(Bcl-2)-associated X protein); HL-60 (human leukaemia cell line); Hepa1c1c7 (hepatoma cell line Hepa1c1c7); DU-145, PC-3, and JCA (prostate cancer cell lines); MCF-7 (human breast cancer cell line); NF-κB (nuclear factor-κB); IL-1β (interleukin-1β); NLRP1 (NOD-like-receptor containing a Pyrin domain 1); HDACs (Histone deacetylases).

**Table 2 ijms-25-03264-t002:** Protective effects of polyphenols extracted from different food matrices: in vivo study.

Type of Polyphenol	Target	Mechanism of Action	Modification Types	Reference
Curcumin	Intestine Crohn’s Disease (CD)	Causes repression of * TGF-β1	Reduction of intestinal fibrotic stricture in Crohn’s disease	[103]
Epigallocatechin gallate	Lung Cancer	Acts as an alternative immune checkpoint inhibitor	Decrement in tumour multiplicity	[104]
Epigallocatechin gallate	Lung Cancer	Modulates Akt, NF-κB, MAP kinases and cell cycle pathways	Reduction in tumour multiplicity, tumour size	[105]
Epigallocatechin gallate	Colon Cancer	Causes decrease in the levels of proinflammatory eicosanoids, prostaglandin E2, and leukotriene B4	Reduction in tumour growth	[106]
Epigallocatechin gallate	Colon Cancer	Causes apoptosis and augmented expression levels of RXR α, β, and γ in the adenocarcinomas	Reduction in tumour growth	[107]
Epigallocatechin gallate	Skin Cancer	Causes inhibition of the proliferation, inhibition of NF-κB activity, IL-1β secretion related with downregulation of NLRP1	Inhibition of melanoma tumour growth	[108]
Epigallocatechin gallate	Skin Cancer	Induces photoprotective effect against acute UVB	Reduction in tumour size and tumour volume	[109]
Epigallocatechin gallate	Prostate Cancer	Causes inhibition of agonist-dependent AR activation and AR-regulated gene transcription	Reduction in tumour growth	[110]
Allicin	Cholangiocarcinoma	Causes reduction in the activity of the PI3K/AKt signalling pathway	Suppression of the growth of human liver bile duct carcinoma	[111]
Epigallocatechin gallate	Breast Cancer	Causes downregulation of miR-25	Reduction in tumour growth	[112]
Epigallocatechin gallate	Breast Cancer	Causes decrease in AKt and increase in Bax/Bcl-2 ratio, comparable to tamoxifen	Reduction in tumour growth	[113]

* Abbreviation: TGF-β1 (Transforming growth factor-β1); Nrf2 (nuclear factor erythroid 2–related factor 2); LC3 (Light Chain 3); NF-kB (Nuclear factor kappa B); NLR1 (NLR Family Pyrin Domain Containing 1); HDACs (Histone deacetylases); AKT (protein kinase B); Bcl-2 (B-cell lymphoma protein 2; BAX (Bcl-2)-associated X protein); HL-60 (human leukaemia cell line); Hepa1c1c7 (hepatoma cell line Hepa1c1c7); DU-145, PC-3, and JCA (prostate cancer cell lines); MCF-7 (human breast cancer cell line); NF-κB (nuclear factor-κB); IL-1β (interleukin-1β); NLRP1 (NOD-like-receptor containing a Pyrin domain 1); HDACs (Histone deacetylases); miR-25 (micro-RNA25).

**Table 3 ijms-25-03264-t003:** Bioavailability Index of the most representative phytochemicals with antioxidant activity, at the intestinal level.

ID	Bioavailability Index (%)
Caffeic acid	13.4
Chlorogenic acid	148
p-Coumaric acid	2.2
3,5-Dicaffeoylquinic acid	-
Ellagic acid	93.4
Ferulic acid	-
Gallic acid	0.68
Protocatechuic acid	2.5
Rosmarinic acid	-
p-Salicylic acid	-
o-Salicylic acid	-
Syringic acid	7.6
Vanillic acid	-
Quercetin	43.2
Taxifolin	61.1
Isoquercetin	86.8
Hyperoside	95.2
Quercitrin	107
Kaempferol	-
Rutin	138
Narcissoside	121
Isorhamnetin-3-O-glucoside	83.3
Catechin	41.9
Epicatechin	15.2
Epicatechin gallate	92.3
Procyanidin A2	104
Procyanidin B1	213
Procyanidin B2	40.1
Procyanidin C1	-
Total flavanols	72
Luteolin	-
Luteolin-7-O-glucoside	92.6
Tangeretin	39.4
Fisetin	-
Phlorizin	153.1
Methyl gallate	97.6

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
