# Peer review of "Oxidative Stress: The Role of Antioxidant Phytochemicals in the Prevention and Treatment of Diseases"

_ijms, 2024, doi:10.3390/ijms25063264_

Round 1

Reviewer 1 Report

Comments and Suggestions for Authors

The work presented by the authors brings to the fore a characteristic of modern man, namely, daily stress. In trying to find the healthiest lifestyle approach, of course, we try all kinds of strategies, including eating vegetables, fruits or supplementing with natural compounds with antioxidant/antiradical action.

The evaluation of the article was based on:

1.     the introduction systematizes data about oxidative stress, as well as the multitude of damages induced in the body by ROS; systematization of the classes of natural constituents that show antioxidant action; creating a supporting diagram;

2.     the results are systematization for an easy understanding of the importance of natural compounds in the management of oxidative stress at the cellular level; the most important polyphenols isolated from food sources with activity tested in vitro for antioxidant activity in different forms of cancer are represented; other important subchapters from the specialized literature research are those related to the bioavailability of natural compounds, including the in vivo studies carried out on them, the role of natural compounds in prevention;

3.     the discussions are presented systematically and focused on the role of polyphenols in health;

4.     the materials systematically present the databases accessed for the purpose of quantifying the results;

5.     the bibliography is supporting.

Author Response

Reviewer 1

The work presented by the authors brings to the fore a characteristic of modern man, namely, daily stress. In trying to find the healthiest lifestyle approach, of course, we try all kinds of strategies, including eating vegetables, fruits or supplementing with natural compounds with antioxidant/antiradical action.

The evaluation of the article was based on:

  1. the introduction systematizes data about oxidative stress, as well as the multitude of damages induced in the body by ROS; systematization of the classes of natural constituents that show antioxidant action; creating a supporting diagram;

Thank you for your positive comment

  1. the results are systematization for an easy understanding of the importance of natural compounds in the management of oxidative stress at the cellular level; the most important polyphenols isolated from food sources with activity tested in vitro for antioxidant activity in different forms of cancer are represented; other important subchapters from the specialized literature research are those related to the bioavailability of natural compounds, including the in vivo studies carried out on them, the role of natural compounds in prevention;

Thank you for your positive comments

  1. the discussions are presented systematically and focused on the role of polyphenols in health;

Thank you for your positive comments

  1. the materials systematically present the databases accessed for the purpose of quantifying the results;

Thank you, I am really gratified from your positive comments, it was not so easy to write a review on this topic trying to be clear to It hasn't been easy to write this review, trying to be clear, elucidate the main points, and be concise at the same time.

  1. the bibliography is supporting

Thank you

Reviewer 2 Report

Comments and Suggestions for Authors

1-In reviewing the manuscript, I noticed that abbreviations used in the table are not accompanied by their full descriptions. It's crucial for readers to have clarity on the meaning of abbreviations to facilitate understanding. I recommend that the author includes a legend or a footnote in the table providing the full descriptions of the abbreviations used. This will enhance the readability and comprehensibility of the table.

2-While I understand the extensive nature of the reference list, it may be challenging to verify the accuracy and relevance of all 148 references. To ensure the quality of the manuscript, I recommend the authors thoroughly review the references to ensure they are all pertinent to the study and accurately cited. Additionally, utilizing reference management software can help maintain consistency and accuracy throughout the manuscript. Please specific the name of software used and mention it inside the acknowledgement. Finally, providing a brief summary or rationale for the inclusion of key references could assist me in assessing their relevance to the study.

3-I have noticed that some sections of the paper exhibit a style or language that raises concerns about potential use of generative AI software. While the use of such tools can sometimes aid in generating content, it's crucial for the transparency and integrity of the research to clearly attribute any AI-generated content and to ensure that it aligns with ethical standards and scientific rigor. I recommend that the authors explicitly disclose any use of generative AI software in the methodology section and provide details on how the generated content was reviewed, validated, and integrated into the paper. Additionally, conducting a thorough review of the manuscript to ensure coherence, accuracy, and originality would help maintain the credibility of the research. If no software were used in improving the content of manuscript then specify inside the methodology. 

4-I suggest including the chemical structures of the phytochemicals discussed in the manuscript. Providing visual representations of these structures can greatly enhance the understanding of their properties and mechanisms of action for readers. Additionally, including the structures would aid in the identification and interpretation of specific compounds referenced throughout the text. I recommend incorporating a separate section or figure(s) dedicated to illustrating the chemical structures of key phytochemicals, accompanied by brief descriptions of their biological activities and potential relevance to the study. This addition would enrich the comprehensiveness and clarity of the manuscript, enhancing its overall quality.

Comments on the Quality of English Language

acceptable

Author Response

Reviewer 2

1-In reviewing the manuscript, I noticed that abbreviations used in the table are not accompanied by their full descriptions. It's crucial for readers to have clarity on the meaning of abbreviations to facilitate understanding. I recommend that the author includes a legend or a footnote in the table providing the full descriptions of the abbreviations used. This will enhance the readability and comprehensibility of the table.

Thank you I fully agree with you, I added a footnote in the table explaining the abbreviation used.

2-While I understand the extensive nature of the reference list, it may be challenging to verify the accuracy and relevance of all 148 references. To ensure the quality of the manuscript, I recommend the authors thoroughly review the references to ensure they are all pertinent to the study and accurately cited. Additionally, utilizing reference management software can help maintain consistency and accuracy throughout the manuscript. Please specific the name of software used and mention it inside the acknowledgement. Finally, providing a brief summary or rationale for the inclusion of key references could assist me in assessing their relevance to the study.

Thank you for the comments, I edited the reference list, I apologize for the inaccuracy in the format, I now modified the style according to the role of the Journal

3-I have noticed that some sections of the paper exhibit a style or language that raises concerns about potential use of generative AI software. While the use of such tools can sometimes aid in generating content, it's crucial for the transparency and integrity of the research to clearly attribute any AI-generated content and to ensure that it aligns with ethical standards and scientific rigor. I recommend that the authors explicitly disclose any use of generative AI software in the methodology section and provide details on how the generated content was reviewed, validated, and integrated into the paper. Additionally, conducting a thorough review of the manuscript to ensure coherence, accuracy, and originality would help maintain the credibility of the research. If no software were used in improving the content of manuscript then specify inside the methodology. 

Sorry for this, I tried to correct some sentence. I want briefly to explain you that sometimes, when citing research from other authors, we try to avoid plagiarism by finding synonyms with the help of a dictionary and translator. However, these synonyms may sometimes be redundant, excessive, or not entirely relevant. It is not our practice to use AI for writing scientific papers.

 4-I suggest including the chemical structures of the phytochemicals discussed in the manuscript. Providing visual representations of these structures can greatly enhance the understanding of their properties and mechanisms of action for readers. Additionally, including the structures would aid in the identification and interpretation of specific compounds referenced throughout the text. I recommend incorporating a separate section or figure(s) dedicated to illustrating the chemical structures of key phytochemicals, accompanied by brief descriptions of their biological activities and potential relevance to the study. This addition would enrich the comprehensiveness and clarity of the manuscript, enhancing its overall quality.

I included figures with chemical structure of the key phytochemicals discussed in this review,

Thank you for this suggestion, I hadn't thought about it, and it was helpful.

Reviewer 3 Report

Comments and Suggestions for Authors

This review article includes the role of antioxidant phytochemicals in the prevention and treatment of diseases. This manuscript (MS) provided a thorough investigation of oxidative stress and inverse relationship of different phytochemicals. However, for publishing into IJMS, some points need to be considered.

It's better to change the title. Illness may be refer to condition caused by diseases. It's better to use diseases/disorders instead of diseases.

Antioxidant phytochemicals. Is a broad term. Need to be more specific in the abstract, as phytochemical has diverse classification. Should need to be more specific in this regard.

It should be good to add some relevant figures.

Introduction is too long to have some redundant stuff. Intro should be more focused on oxidative stress and related mechanisms. Later, in the discussion of the body, the authors need to materialize the points they have mentioned in the introduction.

Comments on the Quality of English Language

Need minor corrections.

Author Response

Reviewer 3

Comments and Suggestions for Authors

This review article includes the role of antioxidant phytochemicals in the prevention and treatment of diseases. This manuscript (MS) provided a thorough investigation of oxidative stress and inverse relationship of different phytochemicals. However, for publishing into IJMS, some points need to be considered.

It's better to change the title. Illness may be refer to condition caused by diseases. It's better to use diseases/disorders instead of diseases.

Thank you I changed it

Antioxidant phytochemicals. Is a broad term. Need to be more specific in the abstract, as phytochemical has diverse classification. Should need to be more specific in this regard.

Thank you I corrected this term and I edit it in the manuscript

It should be good to add some relevant figures.

I added the figures with the chemical structure of the key phytochemicals mentioned in the review

Introduction is too long to have some redundant stuff. Intro should be more focused on oxidative stress and related mechanisms. Later, in the discussion of the body, the authors need to materialize the points they have mentioned in the introduction.

I tried to improve it also considering the feedback from the other three reviewers as well. It is challenging to modify the manuscript when faced with diverse revisions that may not always align. I aimed to strike a balance with my corrections, with the hope of satisfying all of you.

Reviewer 4 Report

Comments and Suggestions for Authors

This is a review article, therefore, it does not need the results, discussion, material, and methods sections. You must classify the information into sections and give an explanation and critical analysis of the information based on coincidences or controversies between authors.

The objective of the manuscript is interesting since it seeks to provide relevant information on the use of antioxidants for human health; However, the authors have neglected certain aspects in the development of their writing that are discussed below. In addition, the English edition is missing

Line 66.  The authors generalize the concept of phytochemicals saying that they are all anti-oxidants.

Line 72-74.  The cited author mentions other functions of secondary metabolites that the authors omit; Are secondary metabolites of leisure in plants? 

Line 82-83 The authors must indicate that the amount of secondary metabolites varies depending on the climate, type of soil, etc.

Line 92, include the reference

Line 106.  Do the authors describe previous results of their research?

Line 111 -114 They must correct the writing of scientific names

Table 1. A mechanism is different from an effect, the mechanism describes how a phenomenon or effect occurs. You should clarify what you mean by types of modification. For example, does it modify inhibition or cause inhibition?

Line 258 What does this acronym (AUC mean?

Line 297 It is suggested that it be another section The data in Table 2 does not correspond to the title of this one, but rather is from Table 1. The authors must be more careful in presenting the information

Table 2 If the authors should include a bioavailability and accessibility table, it would be very interesting; However, the title of Table 2 refers to other types of information.

Line 438 reference?

Line 455 You must mention examples

Author Response

Reviewer 4

Comments and Suggestions for Authors

This is a review article, therefore, it does not need the results, discussion, material, and methods sections. You must classify the information into sections and give an explanation and critical analysis of the information based on coincidences or controversies between authors.

Thank you. I have rectified the format of the review. I understand your point, but in another review I wrote for the MDPI Journal, reviewers were explicitly asked to structure the review content in alignment with the manuscript format. This is the reason why I used that format.

The objective of the manuscript is interesting since it seeks to provide relevant information on the use of antioxidants for human health; However, the authors have neglected certain aspects in the development of their writing that are discussed below. In addition, the English edition is missing

I corrected accordingly

Line 66.  The authors generalize the concept of phytochemicals saying that they are all anti-oxidants.

Thank you I corrected now in the whole manuscript

Line 72-74.  The cited author mentions other functions of secondary metabolites that the authors omit; Are secondary metabolites of leisure in plants? 

I explicated in the text, thank you

Line 82-83 The authors must indicate that the amount of secondary metabolites varies depending on the climate, type of soil, etc.

I added it in the text

Line 92, include the reference,

I did

Line 106.  Do the authors describe previous results of their research?

I improved the sentence to be clearer

Line 111 -114 They must correct the writing of scientific names

I corrected all the scientific name in the manuscript, Sorry for this

Table 1. A mechanism is different from an effect, the mechanism describes how a phenomenon or effect occurs. You should clarify what you mean by types of modification. For example, does it modify inhibition or cause inhibition?

I corrected in the table, thank you

Line 258 What does this acronym (AUC mean?

I explaining it in the text

Line 297 It is suggested that it be another section The data in Table 2 does not correspond to the title of this one, but rather is from Table 1. The authors must be more careful in presenting the information

You are right sorry for the mistake, I corrected it now in the manuscript

Table 2 If the authors should include a bioavailability and accessibility table, it would be very interesting; However, the title of Table 2 refers to other types of information.,

I included another table with bioaccessibility index of the main phytochemicals, and corrected table 2

Line 438 reference? I added

Line 455 You must mention examples

I explained in the text

Round 2

Reviewer 2 Report

Comments and Suggestions for Authors

The manuscript is publishable to my knowledge.

Reviewer 4 Report

Comments and Suggestions for Authors

Research on natural antioxidant products that can be used as adjuvants in the treatment of different diseases, or to prevent them, is interesting. We hope to soon have encouraging news of experimental results from your research, in vitro or in vivo.

Comments on the Quality of English Language

I am not qualified to assess the quality of English in this paper